# Comparative Analysis of Pneumococcal Serotypes for 10 Years (2014–2024) in the Comunidad Valenciana Region, Spain, and How They Are Correlated with PCV13, PCV20, and PCV21

**DOI:** 10.3390/vaccines13101018

**Published:** 2025-09-29

**Authors:** Laura Diab-Casares, Nuria Tormo-Palop, Rafael Medina-González, Sonia Cortés-Badenes, Francisco Javier Hernández-Felices, Violeta Artal-Muñoz, José Luis Martín-Rodríguez, Francisco Roig-Sena, José Manuel Marín, María Dolores Gómez-Ruiz, Francisco José Rodríguez-Nortes, Mariana Lamas-Santángelo, Concepción Gimeno-Cardona, Remedio Guna-Serrano

**Affiliations:** 1Departamento de Microbiología, Consorcio Hospital General Universitario de Valencia, 46014 Valencia, Spain; diab_lau@gva.es (L.D.-C.); tormo_nur@gva.es (N.T.-P.); medina_rafgon@gva.es (R.M.-G.); cortes_sonbad@gva.es (S.C.-B.); hernandez_fra@gva.es (F.J.H.-F.); artal_vio@gva.es (V.A.-M.); martin_jos@gva.es (J.L.M.-R.); rodriguez_fra@gva.es (F.J.R.-N.); lamas_mar@gva.es (M.L.-S.); guna_rem@gva.es (R.G.-S.); 2Facultad de Medicina, Departamento de Microbiología, Universidad de Valencia, 46010 Valencia, Spain; 3Servicio de Vigilancia y Control Epidemiológico, Dirección General de Salud Pública, Conselleria de Sanitat, Comunitat Valenciana, 46010 Valencia, Spain; roig_fco@gva.es (F.R.-S.); jmarin@odec.es (J.M.M.); 4Servicio de Microbiología, Hospital Universitario y Politécnico La Fe, 46026 Valencia, Spain; gomez_mar@gva.es

**Keywords:** Comunidad Valenciana, invasive pneumococcal disease, serotypes, *Streptococcus pneumoniae*, PCV20, PCV21

## Abstract

**Background/Objectives**: This study analyzes the epidemiology of invasive pneumococcal disease (IPD) and the dynamics of *Streptococcus pneumoniae* (SP) serotypes in the Comunidad Valenciana (CV) region, Spain, over a 10-year period (2014–2024), with particular focus on vaccine coverage of PCV13 compared to the newer PCV20 and PCV21 formulations. **Methods**: A total of 2.014 isolates of SP obtained from sterile fluids were included, with available serotype, demographic data, and vaccination status, which were collected from the Epidemiological Surveillance System (AVE) and the Microbiological Surveillance Network of the CV region (RedMIVA). **Results**: Overall vaccination coverage was low (22.4%), with the highest rates observed in children under 10 years (78%) compared to only 16% in those aged 10–64 years and 22% in those over 64. Serotype distribution revealed 120 distinct serotypes, with serotype 8 (17.6%) and serotype 3 (14.7%) being the most frequent. Serotype 8 predominated among unvaccinated individuals, while serotype 3 remained highly prevalent despite inclusion in PCV13. Other relevant serotypes included 22F, 9N, 19A, 6C, and 23A. Temporal analysis showed that serotype 3 has continued to increase in recent years, whereas serotype 8 rose during the pandemic period but has remained stable in the most recent interval, while 19A, 15A, and 11A significantly declined. Among serotypes with <2% incidence, some, such as 4, 24F, and 38, showed upward trends. **Conclusions**: The findings suggest that PCV20 currently provides broad coverage of dominant serotypes, but PCV21 may offer advantages should serotypes like 23A, 9N, or 15A increase further due to serotype replacement. Continuous epidemiological surveillance is essential to guide evidence-based vaccine policy and anticipate future vaccine reformulations.

## 1. Introduction

*Streptococcus pneumoniae* (SP) remains the leading bacterial cause of severe pneumonia and pneumonia-related mortality worldwide. It is an asymptomatic colonizer of the human nasopharynx, especially in children, and can spread to adjacent areas, causing infections such as otitis, aspiration pneumonia, or invading sterile body fluids, resulting in what is known as invasive pneumococcal disease (IPD). IPD can reach mortality rates of up to 20% for sepsis and 50% for meningitis, being more common in individuals under 5 years of age and over 65 [1,2]. In this regard, pneumococcal vaccines are of particular interest, as they are designed to cover the serotypes most frequently associated with severe pneumonia and IPD.

Currently, there are two categories of vaccines: the pneumococcal polysaccharide vaccine (PPV) and the pneumococcal conjugate vaccine (PCV). The former contains 23 different serotypes and is based on SP capsular polysaccharides (PPV23), promoting immunity by stimulating B lymphocytes, which differentiate into plasma cells responsible for antibody production. It induces only serum IgG secretion, not mucosal IgA. Developed and introduced in 1983, it failed to generate adequate immunity in children and older adults [3].

To overcome this limitation, conjugate vaccines were developed [4]. These are based on the linkage of a weak polysaccharide molecule to a strong antigen, such as a protein, generating a T-cell-dependent immune response in addition to B-cell stimulation. This induces immune memory, a robust antibody response, and mucosal immunity [5,6,7]. The first conjugate vaccine was PCV7, introduced in 2000, followed sequentially by PCV10 and PCV13, with the aim of including additional serotypes and reformulating the antigenic content to increase specificity and serotype coverage [6,8].

The most recent vaccines developed are the 15-, 20-, and 21-valent conjugates (PCV15, PCV20, and PCV21). Their approval was based on solid clinical trials evaluating their safety and immunogenicity compared to previous vaccines [9,10,11]. Focusing on PCV20 and PCV21, the main differences lie in their serotype coverage. Both vaccines share the following serotypes: 3, 6A, 6B, 8, 10A, 11A, 12F, 15B, 19A, 19F, 22F, and 33F. Unique to PCV20 are serotypes 1, 4, 5, 7F, 9V, 14, 18C, and 23F, while PCV21 uniquely includes serotypes 9N, 16F, 17F, 20, 23A, 23B, 24F, 31, and 35B [12,13].

PCV20 was approved by the EMA in 2022 and made available in Spain in 2024 [14]. It is also the vaccine funded by the CV region as of July 2024 [15]. According to the Spanish Association of Pediatrics (AEP) Vaccine Committee, PCV20 is expected to replace PCV13 in the vaccination schedule for children born on or after 1 January 2025 [16].

In light of these considerations, the aim of this study was to assess the frequency of common and uncommon serotypes included in PCV20 and PCV21, as well as non-vaccine serotypes with the potential to cause IPD, in the CV region. The study further sought to identify those serotypes that have significantly increased over the past decade (2014–2024), in order to support the preferential use of one vaccine over the other.

## 2. Materials and Methods

### 2.1. Study Design and Data Collection

This is a retrospective study including a sample of 2.014 SP isolates obtained from sterile clinical specimens collected in hospitals of the CV region over a 10-year period (2014–2024). The dataset includes sociodemographic data (sex and age), serotype, and vaccination status.

Of the total sample, 41.5% were women, and 58.5% were men. The mean age of the population was 61.24 years (± 23.18), ranging from 0 to 100 years. When regrouping the sample into three age brackets (excluding 21 cases with age = 124), the distribution is as follows: 5.6% <10 years, 53.3% between 10 and 65 years, and 41.1% ≥65 years.

### 2.2. Sociodemographic and Analytical Data

All case data were obtained from the Epidemiological Surveillance System (“AVE”), which integrates information from three primary sources: outpatient information systems, hospital preventive medicine departments, and the Microbiological Surveillance Network of the CV region (“RedMIVA”) [17]. RedMIVA provides real-time microbiological testing results from hospitals across the CV region. The following variables were collected and analyzed in this study: age, sex, serotype, and vaccination status.

For the data analysis, it was decided to divide the time ranges into three periods according to the COVID-19 pandemic—2014–2019 (before), 2020–2022 (during), and 2023–2024 (after)—in order to avoid the bias introduced during that period, since, as can be observed, the number of IPD cases reported in 2020 and 2021 was markedly lower than in the other years, partly due to non-pharmaceutical interventions and partly to reduced diagnosis of the disease [18,19]. The number of samples of SP responsible for IPD analyzed by this institution per year is shown in the following figure (Figure 1):

### 2.3. Serotyping

While each RedMIVA-affiliated laboratory determines its own bacteriological detection methods, they all adhere to standardized interpretation criteria. Serotyping of pneumococcal isolates was carried out using the Neufeld capsule test at the Microbiology Service of Hospital La Fe in Valencia or at the National Microbiology Centre from 2014 to 2021. From 2022 onwards, serotyping was performed using polymerase chain reaction (PCR) followed by reverse hybridization (S. PneumoStrip, Operon, Immuno & Molecular Diagnostics) at the current reference center for pneumococcal serotyping in CV region, the Consorcio Hospital General Universitario de Valencia.

### 2.4. Statistical Analysis

For categorical variables, frequencies and percentages were calculated. Differences between groups were assessed using Pearson’s chi-square test; for operating with low expected frequencies, the exact global *p*-value of the Chi-square test was used with Monte Carlo simulation (1000 samples). All multiple comparisons were adjusted using the Bonferroni (Bf) and Benjamin–Hochberg (B-H) correction. Effect sizes were reported to support the interpretation of results, using Cramer’s V for categorical variables.

The classification of effect size magnitude based on Cramer’s V was as follows [20]:0.00–0.09: negligible;0.10–0.29: small;0.30–0.49: medium;≥0.50: large.

A significance level of 5% (α = 0.05) was used for all analyses.

## 3. Results

### 3.1. Vaccination Status Overview

The distribution of vaccination status in the sample was as follows: 69.4% (1397) were unvaccinated, 8.2% (166) were unknown, 6.2% (125) had received PCV13, 11.2% (226) had received PPV23, 1.3% (27) had both, and 3.6% (73) were vaccinated but unknown, as shown in the following sample diagram (Figure 2):

There is an association between age and vaccination status (Chi^2^(4) = 549.880, *p* < 0.001 (*p* = 1.08 × 10^−117^), V = 0.394, IC95% [0.333, 0.450]). By age group, the percentage of unvaccinated individuals is 24% among children under 10 years of age and 75–85% for the two groups older than 10 years (*p*-value for multiple group comparisons adjusted by both B-H and Bf: *p* < 0.001 for comparison between 10–64 and <10 and *p* < 0.001 for >64 vs. <10). The percentage vaccinated with PCV13 is 71% in those <10 years and 2–7% in individuals older than 10 years (*p*-value for multiple group comparisons adjusted by both B-H and Bf—*p* < 0.001 for comparison between <10 and 10–64 and *p* < 0.001 for >64 vs. <10). The percentage vaccinated with PPV23 is 20% in those >64 compared with 4–7% in individuals younger than 64 (*p*-value for multiple group comparisons: *p* = 0.002 (B-H) and *p* = 0.005 (Bf) for comparison between >64 and 10–64 and *p* < 0.001 (B-H and Bf) for >64 vs. <10) (Table 1).

### 3.2. Pneumococcal Serotypes

A total of 120 different serotypes were identified in the sample. The most common was serotype 8, accounting for 17.6% of cases, followed by serotype 3 (14.7%). Serotypes 22F and 9N had incidences between 4 and 5%, while serotypes 14, 19A, 6C, and 23A showed incidences between 3 and 4%. The remaining serotypes each represented less than 3% of the cases.

Working only with serotypes with an incidence greater than 2%, we observed an association between age and serotypes (Chi^2^(28) =95.692, *p* < 0.001 (*p* = 1.045 × 10^−9^) (V = 0.155 [0.146, 0.206])) according to the following:

Serotypes by Age (Table 2):Serotype 8 is more common among individuals older than 10 years (particularly in the 10–64 age group) (<10 vs. 10–64: *p* < 0.001 (B-H and Bf); <10 vs. >64: *p* = 0.037 (Bf) and *p* = 0.012 (B-H));Serotype 3 is more frequent in individuals older than 64 years compared to patients aged 10–64 years (>64 vs. 10–64: *p* = 0.033 (B-H and Bf));Serotypes 10A and 23B are more frequent in children under 10 years of age (10A: <10 vs. 10–64: *p* < 0.001 (B-H and Bf); <10 vs. >64: *p* < 0.001 (B-H and Bf); 23B: <10 vs. 10–64: *p* = 0.029 (B-H)/*p* = 0.015 (Bf); <10 vs. >64: *p* = 0.002 (B-H and Bf)).
vaccines-13-01018-t002_Table 2Table 2Association between serotypes and age groups.
AgeTotal<1010–64>64CountN %CountN %CountN %CountN %Serotypes Total1993100.0%112100.0%819100.0%1062100.0%Others *65933.1%5246.4%27833.9%32931.0%835317.7%54.5%18823.0%16015.1%329314.7%1513.4%9912.1%17916.9%22F995.0%54.5%404.9%545.1%9N794.0%00.0%354.3%444.1%14713.6%54.5%212.6%454.2%19A703.5%54.5%222.7%434.0%6C643.2%10.9%212.6%424.0%23A613.1%32.7%192.3%393.7%10A562.8%108.9%242.9%222.1%15A462.3%32.7%182.2%252.4%31442.2%00.0%151.8%292.7%11A432.2%21.8%131.6%282.6%23B402.0%65.4%172.1%171.6%Invalid150.8%00.0%91.1%60.6%Serotypes with an incidence >2% are shown in the table. * Others: includes all serotypes with an incidence <2%.


### 3.3. Serotypes and Vaccination

There is an association between vaccination status and serotype incidence (Chi^2^(42) = 104.528, *p* < 0.001 (*p* = 3.65 × 10^−8^) (V = 0.140 [0.137, 0.198])). The proportion of serotypes with an incidence below 2% (“Others”) was higher among vaccinated individuals compared with unvaccinated individuals (40–50% vs. 30%) (unvaccinated vs. PCV13: *p* < 0.001 (B-H and Bf); unvaccinated vs. PPV23: *p* = 0.003 (Bf) and *p* = 0.002 (B-H)). Serotype 8 was more frequent among unvaccinated individuals compared with those vaccinated with PCV13 or PPV23 (unvaccinated vs. PCV13: *p* = 0.005 (Bf) and *p* = 0.002 (B-H); unvaccinated vs. PPV23: *p* = 0.001 (B-H and Bf)). In contrast, serotype 10A was more frequent in individuals vaccinated with PCV13 compared with the other groups (PCV13 vs. PPV23: *p* < 0.001 (B-H and Bf); PCV13 vs. unvaccinated: *p* = 0.006 (Bf) and *p* = 0.003 (B-H)). (Chi^2^(28) = 88.305, *p* < 0.001 (*p* = 3.65E-8) (V = 0.158* [0.148, 0.214]) (less than 30% of the cells have a count <5; Fisher’s exact test was not required) (Table 3).

*V = 0.158 [0.148, 0.214]; this low value of Cramer’s V indicates that the association between vaccination status and serotype is weak. However, in scenarios involving large contingency tables (as in this case), Cramer’s V tends to yield smaller values. Therefore, although the effect size in this table is not very high, it may still reflect moderate associations between vaccination status and serotype, particularly for the serotypes with higher prevalence.

For greater clarity, the following figure, Figure 3, combines the serotypes representing >2% in relation to vaccine coverage with PCV13, PCV20, and PCV21.

### 3.4. Serotype Trends over Time

Regarding serotype trends, no serotype showed a statistically significant increase in recent years when analyzed year by year. Serotype 8 increased in 2019 and then remained stable, although descriptively it appears to be declining slightly in the most recent years—this trend, however, was not confirmed statistically. Serotype 3 increased in 2022, followed by a downward trend that was also not statistically significant. The remaining serotypes did not show significant evolution over time. It is worth noting that small sample sizes reduce the statistical power of the tests performed (Figure 4).

As shown in Table 4, serotype 8 increased prevalence from 2020 onwards (2020–2022: *p* = 0.001 (B-H and Bf); 2023–2024: *p* = 0.008 (Bf)/*p* = 0.004 (B-H)—both compared with 2014–2019), although no further increase was observed in the last two years. Serotype 3 tended to be more frequent in 2023–2024 compared with before 2019 (*p* = 0.093 (B-H and Bf), <0.1). Serotype 19A decreased in the last two years (*p* = 0.002 and *p* = 0.004 (B-H and Bf), respectively), while serotypes 15A and 11A also declined since 2019 (*p* = 0.035 and *p* = 0.003 (Bf and B-H), respectively). Only after B-H adjustment was serotype 14 found to be less prevalent during the pandemic (1% vs. 6%, *p* = 0.049).

Among the serotypes with an incidence below 2%, the following showed an increase in the last two years (post-pandemic): 4, 38, 24F, and 17F compared with the pre-pandemic period and 15B/C, 12F, and 16F compared with both the pre-pandemic and pandemic periods (Table 5).

## 4. Discussion

This study presents an analysis of the epidemiological distribution of SP serotypes responsible for IPD over the past 10 years, comparing pneumococcal vaccines—particularly PCV13 versus PCV20 and PCV21—with the aim of determining which vaccine would provide the most appropriate coverage in the CV region based on our results.

IPD remains a high-burden disease worldwide and a major public health concern. The introduction of pneumococcal vaccines into immunization programs has led to a marked decline in vaccine-covered serotypes [21,22], as reflected in our results. However, IPD continues to occur at a considerable frequency, largely due to low vaccination coverage; notably, up to 85% of individuals aged >64 years in our cohort were unvaccinated. Regarding PCVs, PCV13 includes seven fewer serotypes than PCV20 and eight fewer than PCV21. The inclusion of serotype 8 in the newer vaccines is especially noteworthy, as it has been the most frequently isolated serotype globally over the last 7 years, both according to our data from the CV region and ECDC reports across Europe over the past 6 years [23], in both vaccinated and unvaccinated individuals. This trend reflects an indirect effect of introducing PCV13 into the regional vaccination schedule in 2015 [24], the same year serotype 8 (not included in PCV13) began rising progressively, becoming the most prevalent. This occurred despite being included in PPV23, whose effectiveness against serotype 8 has been reported at 46% in a study conducted in Germany [25]. The lack of protection from PPV23 and the absence of PCV13 likely contributed to the dominance of serotype 8. Furthermore, its increased presence in Spain from 2015 to 2018 has been linked to the expansion of a single clone (ST53) [26], which was also demonstrated by our research group in another study using whole-genome sequencing (WGS), confirming ST53 as the dominant clone within serotype 8 in our setting [27].

Serotype 3 is more prevalent in children under 10 and adults over 64 and is the second most frequent in individuals aged 10–64, despite being included in PCV13. This is particularly concerning, as it is one of the most invasive serotypes and associated with older patients, comorbidities, and higher lethality [28]. The lack of PCV13 efficacy against serotype 3 in children is likely a key reason for its persistent incidence [29]. Some of the reasons that confer serotype 3 with high evasiveness to PCV13 include, on the one hand, its thick and viscous polysaccharide capsule, which hinders antibody access. In addition, the linkage between the capsule and the bacterial surface is not covalent, allowing the bacterium to shed capsule material bound to antibodies. On the other hand, the bacterial surface carries negative charges, which may repel immune cells [30]. For these reasons, the inclusion of serotype 3 in both PCV20 and PCV21 has received considerable attention. A Phase 3 clinical trial by Essink et al. demonstrated that PCV20 elicited a non-inferior immunological response to PCV13 for this serotype, reaching adequate opsonophagocytic activity (OPA) across adult age groups [10]. PCV21 also showed robust immunogenicity against serotype 3, in some cases equal to or superior to PCV20 [31].

Serotype 23A, which has increased over the past three years in the region, without reaching statistical significance, is another serotype of epidemiological concern—only covered by PCV21. A study by Løchen et al. identified divergent serotype trends across regions yet consistently reported a progressive rise in 23A, which was attributed to the replacement of non-PCV13 serotypes [32]. Although not highly virulent, its persistent increase could warrant future vaccine inclusion and closer surveillance.

Continuing along serotypes with >2% incidence, serotypes 19A, 15A and 11A have shown a statistically significant decrease, while 22F, 9N, 31 and 23B have also declined, though not significantly. In contrast, serotypes 14, 6C, and 10A have remained stable over the past decade, excluding the pandemic years, although with non-negligible incidences. Notably, 9N, 15A, 31, and 23B would no longer be covered if switching from PCV21 to PCV20. Conversely, serotype 14 is included in PCV20 but not in PCV21. Serotype 6C is not covered by any of the current vaccines, and its incidence in IPD in our study is around 3.3% in CV, a finding that supports considering its future inclusion in higher-valent conjugate vaccines, particularly if its prevalence continues to rise. Serotypes 19A and 11A have shown low attack rates and have decreased in incidence in recent years, likely as a response to immunization with PPV23 and PCV13.

Among serotypes with an incidence below 2% that have increased in recent years, only serotype 4 is included in PCV20, while serotype 24F would be covered exclusively by PCV21. Serotype 38 is not targeted by either vaccine. According to the 2023 report of the Spanish National Epidemiological Surveillance Network (RENAVE), the serotypes with increasing incidence were 22F, 38, 4, 12F, and 3. [33]. Notably, serotype 4 (included only in PCV20) should be monitored closely, as it has been responsible for recent occupational outbreaks in shipyard workers [34].

With respect to study limitations, the 10-year period encompasses data up to 2024. A reassessment in the near future will be essential to determine whether serotypes excluded from PCV21 but currently showing an upward trend continue to increase, potentially as a consequence of serotype replacement under PCV20 use. In addition, expanding the sample size would strengthen statistical power to ascertain whether increases in low-prevalence serotypes (<2%) are of statistical significance. Moreover, this article limits its scope to describing the potential serotype coverage of the vaccine formulation, without adjusting for clinical conditions (e.g., comorbidities) or clinical characteristics (such as manifestations and outcomes).

## 5. Conclusions

PCV20 covers the main circulating serotypes currently, including 8, 3, 22F, 14, 19A, 11A, and 10A. PCV21 could be superior if serotypes such as 9N, 23A/B, 15A, and 31 increase in incidence as a consequence of serotype replacement following the introduction of PCV20. There are two serotypes, 6C and 38, showing a certain incidence with an increase in recent years that are not covered by any of the currently available vaccines, which should, therefore, be closely monitored. This study provides preliminary data that must be confirmed with further follow-up. At present, the rise in serotypes not included in PCV20 appears residual. It remains to be seen what collateral impact PCV20 inclusion in vaccination programs may have. 

Future research should prioritize the development and reformulation of vaccines to enhance immune responses against serotype 3, one of the most lethal and evasive serotypes not well controlled by currently available pneumococcal conjugate vaccines.

This study underscores the importance of ongoing epidemiological surveillance of IPD to evaluate the evolution of vaccine and non-vaccine serotypes, adapt vaccination coverage to the local epidemiology, and guide future evidence-based vaccine development. In addition, these findings highlight the need to strengthen vaccination campaigns and educational efforts, particularly for individuals over 64 years of age, as well as to promote greater awareness of the importance of vaccination.

## Figures and Tables

**Figure 1 vaccines-13-01018-f001:**
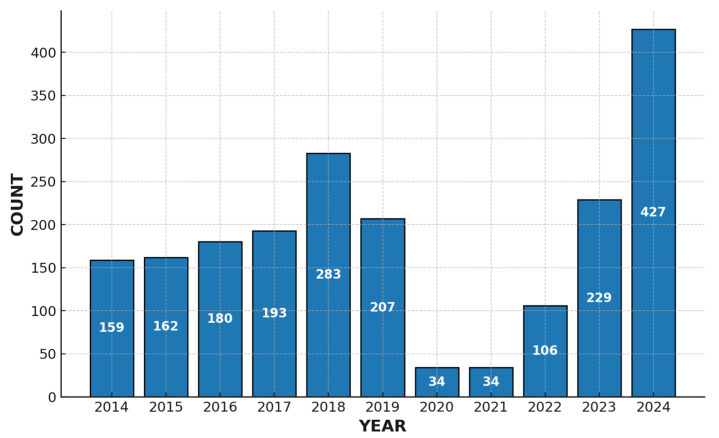
Distribution of IPD cases by year.

**Figure 2 vaccines-13-01018-f002:**
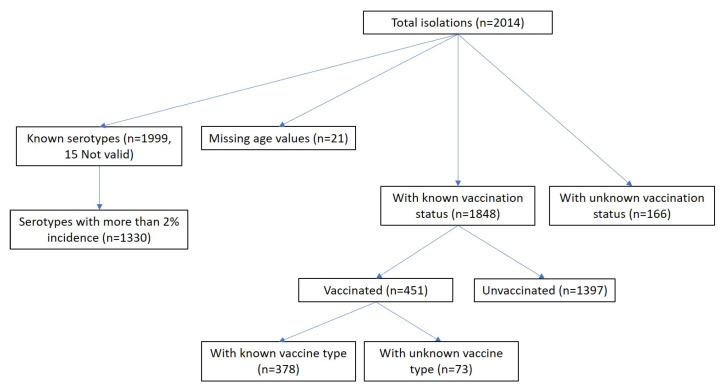
Sample diagram.

**Figure 3 vaccines-13-01018-f003:**
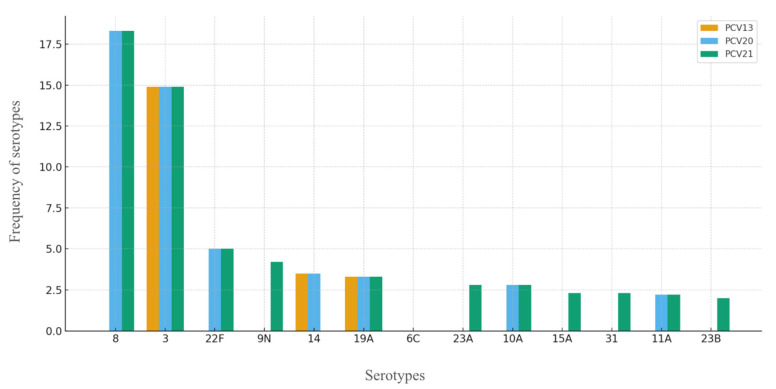
Coverage of serotypes by PCV13, PCV20, and PCV21.

**Figure 4 vaccines-13-01018-f004:**
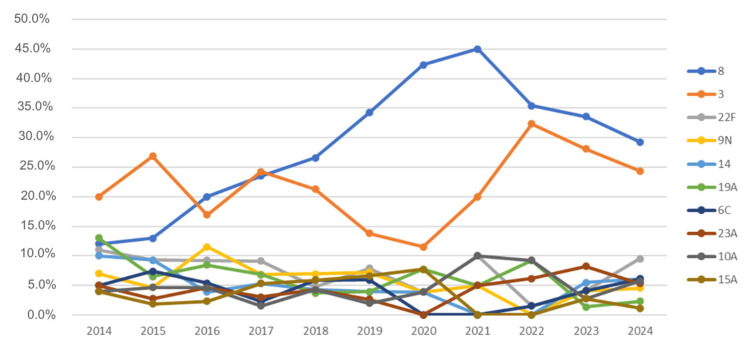
Incidence of IPD cases by responsible serotype.

**Table 1 vaccines-13-01018-t001:** Vaccination rates by age group and type of pneumococcal vaccine.

	Age							
	Total		<10		10–64		>64	
	Count	N %	Count	N %	Count	N %	Count	N %
Total *	1775	100.0%	75	100.0%	729	100.0%	971	100.0%
Not vaccinated	1397	78.7%	18	24.0%	625	85.7%	754	77.7%
Vaccinated with PCV13	125	7.0%	53	70.7%	51	7.0%	21	2.2%
Vaccinated with PPV23	226	12.7%	3	4.0%	45	6.2%	178	18.3%
Vaccinated with both PCV13 and PPV23	27	1.5%	1	1.3%	8	1.1%	18	1.9%

* Groups with unknown vaccination status or unknown vaccine were excluded from the analysis.

**Table 3 vaccines-13-01018-t003:** Serotypes with an incidence greater than 2% in relation to PCV13 and PPV23 vaccination status.

		Vaccination Status									
		Total		Not Vaccinated		Vaccinated with PCV13		Vaccinated with PPV2323		Vaccinated with PCV13 and PPV23	
		Count	N %	Count	N %	Count	N %	Count	N %	Count	N %
Serotypes	Total	1775	100.0%	1397	100.0%	125	100.0%	226	100.0%	27	100.0%
	Others *	575	32.4%	413	29.6%	61	48.8%	87	38.5%	14	51.9%
	8	325	18.3%	286	20.5%	11	8.8%	23	10.2%	5	18.5%
	3	265	14.9%	218	15.6%	12	9.6%	32	14.2%	3	11.1%
	22F	89	5.0%	74	5.3%	5	4.0%	8	3.5%	2	7.4%
	9N	74	4.2%	66	4.7%	0	0.0%	8	3.5%	0	0.0%
	14	63	3.5%	53	3.8%	3	2.4%	7	3.1%	0	0.0%
	19A	59	3.3%	48	3.4%	5	4.0%	6	2.7%	0	0.0%
	6C	59	3.3%	44	3.1%	2	1.6%	13	5.8%	0	0.0%
	23A	50	2.8%	39	2.8%	1	0.8%	9	4.0%	1	3.7%
	10A	50	2.8%	32	2.3%	12	9.6%	5	2.2%	1	3.7%
	15A	41	2.3%	29	2.1%	6	4.8%	6	2.7%	0	0.0%
	31	41	2.3%	34	2.4%	0	0.0%	7	3.1%	0	0.0%
	11A	39	2.2%	26	1.9%	4	3.2%	9	4.0%	0	0.0%
	23B	35	2.0%	27	1.9%	3	2.4%	5	2.2%	0	0.0%
	Invalidated	10	0.6%	8	0.6%	0	0.0%	1	0.4%	1	3.7%

* Others: includes all serotypes with an incidence <2%. Groups with unknown vaccination status or unknown vaccine were excluded from the analysis.

**Table 4 vaccines-13-01018-t004:** Grouped-year periods and incidence of IPD-causing serotypes.

		Year							
		Total		2014–2019		2020–2022		2023–2024	
		Count	N %	Count	N %	Count	N %	Count	N %
Serotype	Total	2014	100.0%	1184	100.0%	174	100.0%	656	100.0%
	Others *	669	33.2%	374	31.6%	63	36.2%	232	35.4%
	8	354	17.6%	185	15.6%	43	24.7%	126	19.2%
	3	297	14.7%	164	13.9%	28	16.1%	105	16.0%
	22F	101	5.0%	66	5.6%	4	2.3%	31	4.7%
	9N	80	4.0%	60	5.1%	2	1.1%	18	2.7%
	14	71	3.5%	46	3.9%	1	0.6%	24	3.7%
	19A	70	3.5%	53	4.5%	9	5.2%	8	1.2%
	6C	66	3.3%	43	3.6%	1	0.6%	22	3.4%
	23A	61	3.0%	30	2.5%	5	2.9%	26	4.0%
	10A	56	2.8%	28	2.4%	9	5.2%	19	2.9%
	15A	46	2.3%	37	3.1%	2	1.1%	7	1.1%
	31	45	2.2%	32	2.7%	3	1.7%	10	1.5%
	11A	43	2.1%	37	3.1%	2	1.1%	4	0.6%
	23B	40	2.0%	29	2.4%	2	1.1%	9	1.4%
	Invalid	15	0.7%	0	0.0%	0	0.0%	15	2.3%

Serotypes with an incidence >2% are shown in the table. * Others: includes all serotypes with an incidence <2%.

**Table 5 vaccines-13-01018-t005:** IPD-causing serotypes by year groupings with less than 2% incidence.

		Year							
		Total		2014–2019		2020–2022		2023–2024	
		Count	N %	Count	N %	Count	N %	Count	N %
Serotype	Total	2014	100.0%	1184	100.0%	174	100.0%	656	100.0%
	4	29	1.4%	10	0.8%	3	1.7%	16	2.4%
	16F	29	1.4%	29	2.4%	0	0.0%	0	0.0%
	38	23	1.1%	9	0.8%	0	0.0%	14	2.1%
	12F	22	1.1%	4	0.3%	5	2.9%	13	2.0%
	17F	12	0.6%	10	0.8%	2	1.1%	0	0.0%
	24F	11	0.5%	1	0.1%	0	0.0%	10	1.5%
	15B/C	10	0.5%	1	0.1%	4	2.3%	5	0.8%

Only serotypes with an incidence below 2% that show differences between periods are presented. The reference year is 2014.

## Data Availability

Data are contained within this article.

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
