# Peer review of "Comparative Analysis of Pneumococcal Serotypes for 10 Years (2014–2024) in the Comunidad Valenciana Region, Spain, and How They Are Correlated with PCV13, PCV20, and PCV21"

_vaccines, 2025, doi:10.3390/vaccines13101018_

Round 1

Reviewer 1 Report

Comments and Suggestions for Authors

Diab-Casares et al. conducted a retrospective study on 2,000 IPD cases in Spain (Comunidad Valenciana), based on the local Epidemiological Surveillance System.

Aim of the study was to evaluate the potential coverage of the currently available pneumococcal conjugate vaccines—PCV13, PCV20, and the newer PCV21—in order to inform vaccine policy decisions.

The study shows that serotype 8 (covered by PCV20 and PCV21) and serotype 3 (covered by PCV20, PCV21 and PCV13) are the most represented serotypes, in line with scientific literature.

Based on these findings, the authors argue that PCV20 currently provides a broad coverage of the most prevalent serotypes, but the new PCV21 formulation could offer an advantage in the future if serotypes such as 9N, 23A or 15A (not covered by PCV20) continue to rise as a consequence of  serotype replacement under vaccination coverage pressure.

Please consider the following suggestions:

  • From the introduction “To overcome this limitation, conjugate vaccines were developed”: I would support this statement with relevant references (i.e. doi: 10.1038/d41573-023-00119-4  and  doi: 10.7416/ai.2024.2614
  • In the methods section authors mention comorbidities as variable of interest but comorbidities are not described along the paper.
  • COVID-19 effect on IPD incidence (Figure 2) was not commented.
  • With reference to table 4: as grouped-years periods are unequally distributed, it is not possible to affirm that “15A and 11A also showed reduced presence since 2019”. Please consider a different grouping criteria or justify the actual one it in the methods.
  • Table 5 caption “IPD-causing serotypes grouped into three time periods”: apparently, time periods are only two in the table (2014-2021 vs 2022-2024) and the grouping criteria has changed from tale 4: please consider to make consistent table 4 with table 5 or justify your choice in the methods section.
  • As the focus of the entire article is on the comparison between PCV13, PCV20 and PCV21 serotype coverage, it would be useful to have a graphical representation, at least for the most represented serotypes.  
  • With reference to study limitations, it would be useful to specify that the paper only describe potential serotypes’ coverage by vaccine formulation, without any correction for clinical condition (comorbidities) or clinical features (clinical manifestation and outcome): for instance, it has been stated that among PCV13 vaccinated ones serotype 3 is the most prevalent serotype; however, serotype 3 is the most prevalent in the >64 years group, which may be more frequently associated to comorbidities.

Minor points:

  • Row 170  contains a typo: “Table 2014. period (Chi²(12) = 58.456, p < 0.001, Cramer’s V = 0.210, small effect size).”

Author Response

Dear Editor,

We would like to thank you for these valuable comments and suggestions, which have greatly helped improve our manuscript. We have carefully reviewed them and provided point-by-point answers under each.

Please find the responses below.

Kind regards.

Reviewer(s)' Comments to Author:

Reviewer: 1

Diab-Casares et al. conducted a retrospective study on 2,000 IPD cases in Spain (Comunidad Valenciana), based on the local Epidemiological Surveillance System.

Aim of the study was to evaluate the potential coverage of the currently available pneumococcal conjugate vaccines—PCV13, PCV20, and the newer PCV21—in order to inform vaccine policy decisions.

The study shows that serotype 8 (covered by PCV20 and PCV21) and serotype 3 (covered by PCV20, PCV21 and PCV13) are the most represented serotypes, in line with scientific literature.

Based on these findings, the authors argue that PCV20 currently provides a broad coverage of the most prevalent serotypes, but the new PCV21 formulation could offer an advantage in the future if serotypes such as 9N, 23A or 15A (not covered by PCV20) continue to rise as a consequence of  serotype replacement under vaccination coverage pressure.

Please consider the following suggestions:

  • From the introduction “To overcome this limitation, conjugate vaccines were developed”: I would support this statement with relevant references (i.e. doi: 10.1038/d41573-023-00119-4  and  doi: 10.7416/ai.2024.2614

Response: Thank you for the comment. We have added it.

  • In the methods section authors mention comorbidities as variable of interest but comorbidities are not described along the paper.

Response: Thank you for the comment. We have removed it.

  • COVID-19 effect on IPD incidence (Figure 2) was not commented.

Response: Thank you for the comment. We have commented it.

  • With reference to table 4: as grouped-years periods are unequally distributed, it is not possible to affirm that “15A and 11A also showed reduced presence since 2019”. Please consider a different grouping criteria or justify the actual one it in the methods.

Response: We appreciate the comment. The grouping criterion was pre-, during, and post-pandemic, since it was a key event in the topic addressed in the article. Moreover, when working with percentages, it is valid to compare non-equivalent time periods, as percentages, by being relative to N, eliminate any possible discrepancies in the sample sizes of the groups and make them comparable.

  • Table 5 caption “IPD-causing serotypes grouped into three time periods”: apparently, time periods are only two in the table (2014-2021 vs 2022-2024) and the grouping criteria has changed from tale 4: please consider to make consistent table 4 with table 5 or justify your choice in the methods section.

Response: Thank you for the comment. We have corrected.

  • As the focus of the entire article is on the comparison between PCV13, PCV20 and PCV21 serotype coverage, it would be useful to have a graphical representation, at least for the most represented serotypes.  

Response: Thank you for the comment. We added it.

  • With reference to study limitations, it would be useful to specify that the paper only describe potential serotypes’ coverage by vaccine formulation, without any correction for clinical condition (comorbidities) or clinical features (clinical manifestation and outcome): for instance, it has been stated that among PCV13 vaccinated ones serotype 3 is the most prevalent serotype; however, serotype 3 is the most prevalent in the >64 years group, which may be more frequently associated to comorbidities.

Response: Thank you for the comment. We have added it.

Minor points:

  • Row 170  contains a typo: “Table 2014. period (Chi²(12) = 58.456, p < 0.001, Cramer’s V = 0.210, small effect size).”

Response: Thank you for the comment. We have deleted it.

Reviewer 2 Report

Comments and Suggestions for Authors

The paper aims to study the epidemiology of invasive pneumococcal 18 disease (IPD) and Streptococcus pneumoniae serotype dynamics in the Comunidad 19 Valenciana (CV), Spain, for a 10-year (2014–2024) interval with special reference to vac-20 cine coverage and efficacy of PCV13 against the latest PCV20 and PCV21 for-21 mulations. Although the article shows some scientific interest, there are some noteworthy observations:

  • The title is to be improved to "Comparative analysis of pneumococcal serotypes for 10 years (2014–2024) in the Comunidad Valenciana region, Spain and how they are correlated with PCV13, PCV20 and PCV21".
  • The manuscript references a total sample of 2,014 isolates, yet Table 1 reports Total = 1993, Table 2 uses 1319, Table 4 uses 1330, and you instruct to exclude "21 cases with age = 124". These inconsistencies are unsettling and make the results unverifiable. Include explicit consort-style flowchart list: total isolates, exclusions (reasons and number), number with known serotype, number with known vaccine status, number used in each analysis.
  • It is extreme missingness and misinterpretation, since 77.4% are "Not vaccinated/Unknown" (i.e., most cases have unknown status). You group "unvaccinated/unknown" together for analyses — this will bias any conclusion regarding vaccine efficacy or association with serotype.
  • The reasoning that persistence of serotype 3 "reflects limited vaccine effectiveness" over-interprets cross-sectional case counts. You can't arrive at vaccine effectiveness (VE) from the proportion of isolates with no denominators or a VE design. To fix this, you need to: (a) split unknown and confirmed unvaccinated into tables/analyses; (b) perform sensitivity analyses restricted to cases with known vaccination status; (c) avoid causal language about vaccine effectiveness unless you do an accepted VE design.
  • RedMIVA/hospital surveillance participation, clinical testing behavior, or laboratory practices can change over time (especially during COVID). The case decline in 2020–2021 likely captures pandemic effects on healthcare utilization and testing — this can distort analysis of trends. Kindly, report surveillance network stability (participating labs annually, volumes tested). When feasible, adjust analyses to control for testing/coverage change; at least address this limitation a priori and report annual laboratory participation or specimens tested.
  • Manuscript mentions WGS and clonality (ST53) but omits methods for serotyping (Quellung, PCR, WGS?) or a sampling strategy for WGS (how many genomes, how they are selected, accession numbers). Reproducibility is lost. Add a full Methods subsection: serotyping assay(s), quality control, proportion of isolates typed by each method, WGS methods (library/prep, sequencing platform, assembly/MLST pipeline), and deposit sequences (ENA/NCBI) with accession numbers.
  • You're reporting lots of chi-square tests across lots of serotypes × years × vaccine categories. Chi-square assumptions. (Version ≥Version ≥5) are likely to be violated for. most cells. (Most serotypes. have low counts). You sometimes report Fisher exact tests for 2×2 but where is unclear.
  • You also mention Bonferroni correction, but you don't report the. number. of comparisons or show adjusted p-values. Bonferroni is too stringent for all tests and might damp down important signals; false discovery rate (Benjamini–Hochberg) would be preferable. Improve on this by: (a) showing how many tests were adjusted and reporting raw and adjusted p-values in supplementary tables; (b) if cell counts are small, utilize Fisher's exact or exact Monte-Carlo p-values or rarify rare serotypes; (c) FDR control instead of Bonferroni would be suitable for exploratory surveillance.
  • You state "no serotype showed a significant increase year-by-year" and then use grouped-year analyses with non-uniform group lengths (e.g., 2014–2019 vs 2020–2022 vs 2023–2024; elsewhere 2014–2021 vs 2022–2024). Grouping seems arbitrary and could alter results.

Author Response

Dear Editor,

We would like to thank you for these valuable comments and suggestions, which have greatly helped improve our manuscript. We have carefully reviewed them and provided point-by-point answers under each.

Please find the responses below.

Kind regards.

Reviewer(s)' Comments to Author:

Reviewer: 2

The paper aims to study the epidemiology of invasive pneumococcal 18 disease (IPD) and Streptococcus pneumoniae serotype dynamics in the Comunidad 19 Valenciana (CV), Spain, for a 10-year (2014–2024) interval with special reference to vac-20 cine coverage and efficacy of PCV13 against the latest PCV20 and PCV21 for-21 mulations. Although the article shows some scientific interest, there are some noteworthy observations:

  • The title is to be improved to "Comparative analysis of pneumococcal serotypes for 10 years (2014–2024) in the Comunidad Valenciana region, Spain and how they are correlated with PCV13, PCV20 and PCV21".

Response: Thank you for the comment. We have modified it.

  • The manuscript references a total sample of 2,014 isolates, yet Table 1 reports Total = 1993, Table 2 uses 1319, Table 4 uses 1330, and you instruct to exclude "21 cases with age = 124". These inconsistencies are unsettling and make the results unverifiable. Include explicit consort-style flowchart list: total isolates, exclusions (reasons and number), number with known serotype, number with known vaccine status, number used in each analysis.

Response: Thank you for the comment. Table 1: n=1993 cases because there are 21 missing values for age (2014–21=1993); Table 2: analysis will be done with serotypes with more than 2% incidence (n=1330) minus the 11 missing values for age in those serotypes (n=1319); Table 4–5: analysis will be done with serotypes with more than 2% incidence (n=1330); Table 6: analysis will be done with the entire sample (n=2014).

  • It is extreme missingness and misinterpretation, since 77.4% are "Not vaccinated/Unknown" (i.e., most cases have unknown status). You group "unvaccinated/unknown" together for analyses — this will bias any conclusion regarding vaccine efficacy or association with serotype.

Response: Thank you for the comment. Your comment made us reflect and, although at first it seemed like a good idea to group them into 2 ranges, it is true that the information is clearer and more informative when separated into 3. Therefore, we have made this change.

  • The reasoning that persistence of serotype 3 "reflects limited vaccine effectiveness" over-interprets cross-sectional case counts. You can't arrive at vaccine effectiveness (VE) from the proportion of isolates with no denominators or a VE design. To fix this, you need to: (a) split unknown and confirmed unvaccinated into tables/analyses; (b) perform sensitivity analyses restricted to cases with known vaccination status; (c) avoid causal language about vaccine effectiveness unless you do an accepted VE design.

Response: Thank you for your comment. You are right, since our study is not a clinical trial assessing vaccine effectiveness, we have corrected it to avoid any misinterpretation.

  • RedMIVA/hospital surveillance participation, clinical testing behavior, or laboratory practices can change over time (especially during COVID). The case decline in 2020–2021 likely captures pandemic effects on healthcare utilization and testing — this can distort analysis of trends. Kindly, report surveillance network stability (participating labs annually, volumes tested). When feasible, adjust analyses to control for testing/coverage change; at least address this limitation a priori and report annual laboratory participation or specimens tested.

Response: Thank you for the comment. We have added it.

  • Manuscript mentions WGS and clonality (ST53) but omits methods for serotyping (Quellung, PCR, WGS?) or a sampling strategy for WGS (how many genomes, how they are selected, accession numbers). Reproducibility is lost. Add a full Methods subsection: serotyping assay(s), quality control, proportion of isolates typed by each method, WGS methods (library/prep, sequencing platform, assembly/MLST pipeline), and deposit sequences (ENA/NCBI) with accession numbers.

Response: Thank you for the comment. We have added the serotyping information. With regard to sequencing, we did not add anything because it was not used in this study. I only mentioned it in the discussion, since it was performed in another study we conducted. However, I do not find it appropriate to include an explanation of the sequencing methodology here, as it was not applied in the present work.

  • You're reporting lots of chi-square tests across lots of serotypes × years × vaccine categories. Chi-square assumptions. (Version ≥Version ≥5) are likely to be violated for. most cells. (Most serotypes. have low counts). You sometimes report Fisher exact tests for 2×2 but where is unclear.

Response: Thank you for the comment. This is because there are no 2×2 tables in your study, and the chi² tests already involve multiple post-hoc comparisons.

  • You also mention Bonferroni correction, but you don't report the. number. of comparisons or show adjusted p-values. Bonferroni is too stringent for all tests and might damp down important signals; false discovery rate (Benjamini–Hochberg) would be preferable. Improve on this by: (a) showing how many tests were adjusted and reporting raw and adjusted p-values in supplementary tables; (b) if cell counts are small, utilize Fisher's exact or exact Monte-Carlo p-values or rarify rare serotypes; (c) FDR control instead of Bonferroni would be suitable for exploratory surveillance.

Response: Thank you for the comment. To address this limitation, we compared Bonferroni with Benjamini–Hochberg, and additionally reported the exact p-value using the Chi² test with Monte Carlo simulation (1,000 samples), as well as the post-hoc p-values corrected by both Bonferroni and Benjamini–Hochberg.

  • You state "no serotype showed a significant increase year-by-year" and then use grouped-year analyses with non-uniform group lengths (e.g., 2014–2019 vs 2020–2022 vs 2023–2024; elsewhere 2014–2021 vs 2022–2024). Grouping seems arbitrary and could alter results.

Response: Thank you for the comment. We have ultimately revised it into three time periods defined by the COVID era.

Reviewer 3 Report

Comments and Suggestions for Authors

Thanks for inviting me to review this manuscript. Overall, it is very interesting, it and provides data about thousands of patients. I have some comments that could be of use:

  1. Regarding statistical analysis, the reviewers mention using Cramer's V to measure effect size. It could be a good idea to include a more detailed interpretation of these values, particularly the "small effect size" noted for the relationship between serotype and vaccination status, to better bring the findings into context for the reader.
  2. The discussion could be strengthened by commenting more on serotype 3 evasiveness. Perhaps the authors could elaborate further on the biological reasons for this lack of efficacy
  3. The discussion could be expanded to include some specific recommendations for what to monitor in the future. For example, are there any non-PCV serotypes that, while currently at low incidence, show an upward trend and should be closely watched?
  4. Since serotype 8 is among the most common, and PCV20 probably covers serotype 8, but did not reach statistical significance, could the authors comment on that?
  5. The conlcusions section is written with bullet points. I think it would be better to be in a single paragraph
  6. Minor typographical errors and grammatical inconsistencies could be corrected to improve the overall quality of the manuscript

Author Response

Dear Editor,

We would like to thank you for these valuable comments and suggestions, which have greatly helped improve our manuscript. We have carefully reviewed them and provided point-by-point answers under each.

Please find the responses below.

Kind regards.

Reviewer(s)' Comments to Author:

Reviewer: 3

Thanks for inviting me to review this manuscript. Overall, it is very interesting, it and provides data about thousands of patients. I have some comments that could be of use:

  1. Regarding statistical analysis, the reviewers mention using Cramer's V to measure effect size. It could be a good idea to include a more detailed interpretation of these values, particularly the "small effect size" noted for the relationship between serotype and vaccination status, to better bring the findings into context for the reader.

Response: Thank you for the comment. We have added it.

  1. The discussion could be strengthened by commenting more on serotype 3 evasiveness. Perhaps the authors could elaborate further on the biological reasons for this lack of efficacy

Response: Thank you for the comment. We have added it.

  1. The discussion could be expanded to include some specific recommendations for what to monitor in the future. For example, are there any non-PCV serotypes that, while currently at low incidence, show an upward trend and should be closely watched?

Response: Thank you for the comment. We have added it.

  1. Since serotype 8 is among the most common, and PCV20 probably covers serotype 8, but did not reach statistical significance, could the authors comment on that?

Response: Thank you for your comment. Indeed, serotype 8 remains the most frequent; however, when we state that it did not reach statistical significance, we are referring to the fact that it has not shown a significant increase over the past ten years but has rather remained stable while continuing to be the most prevalent serotype. This highlights the importance of its inclusion in the new PCV20 and PCV21 vaccines.

  1. The conlcusions section is written with bullet points. I think it would be better to be in a single paragraph.

Response: Thank you for the comment. We have modified it.

  1. Minor typographical errors and grammatical inconsistencies could be corrected to improve the overall quality of the manuscript.

Response: Thank you for the comment. We have corrected it.

Round 2

Reviewer 2 Report

Comments and Suggestions for Authors

After thoroughly reviewing the revised manuscript and considering the authors' revisions and responses to the referee's comments, I find that the manuscript has been significantly improved. The authors have effectively addressed the concerns, thereby enhancing the clarity and scientific rigour of their study. The revisions have clarified the methodology, improved the presentation of results, and strengthened the discussion and conclusions.

Therefore, I believe that the manuscript now meets the standards required for publication in Vaccines, and I recommend that it be accepted for publication.

Reviewer 3 Report

Comments and Suggestions for Authors

The manuscript has been improved